# Moderators of Change in Physical Activity Levels during Restrictions Due to COVID-19 Pandemic in Young Urban Adults

**Josip Karuc [1,\*], Maroje Sorić [1,2], Ivan Radman [1]**  **and Marjeta Mišigoj-Duraković [1]**

[1]    Faculty of Kinesiology, University of Zagreb, 10000 Zagreb, Croatia; maroje.soric@kif.unizg.hr (M.S.); ivan.radman@kif.unizg.hr (I.R.); marjeta.misigoj-durakovic@kif.unizg.hr (M.M.-D.)
[2]    Faculty of Sport, University of Ljubljana, 1000 Ljubljana, Slovenia
\*    Correspondence: josip.karuc@kif.unizg.hr; Tel.: +385-9-1582-3504

**Abstract:** This study aimed to investigate moderators of change in physical activity (PA) levels after 30 days (30-d) of restrictions due to the COVID-19 pandemic in young adults. This research is an extension of the CRO-PALS study and analyses for this study were performed on young adults (20–21 y.o., n = 91). Moderate-to-vigorous physical activity (MVPA), sport participation, student and socioeconomic status were assessed pre- and post-30-d restrictions. Differences in MVPA levels were examined using repeated-measures ANOVAs. After 30-d of restrictions, the drop in MVPA in females ($-64.8$ min/day, $p = 0.006$) and males was shown ($-57.7$ min/day, $p < 0.00$). However, active participants decreased, while non-active peers increased their MVPA level ($-100.7$ min/day, $p < 0.00$, and $+48.9$ min/day, $p = 0.051$, respectively). Moreover, students and non-students decreased their MVPA level ($-69.0$ min/day, $p < 0.00$, and $-35.0$ min/day, $p = 0.22$, respectively) as well as sport participants and non-sport participants ($-95.3$ min/day, $p < 0.001$, and $-53.9$ min/day, $p < 0.00$, respectively). Our results suggest that 30-d of restrictions equally affect females and males where the evident drop in MVPA is seen in both genders. However, active people decreased their PA level during lockdown and the opposite pattern was seen in non-active peers, where restrictions for them can represent an opportunity to change their behavior in a positive direction in order to gain better health status.

**Keywords:** coronavirus; cardiovascular health; quarantine; lockdown

## 1. Introduction

The COVID-19 pandemic currently represents a major global problem, with significant health, social and economic consequences. Indeed, the coronavirus has infected more than 15 million people, and more than half a million had died from the new disease by the end of 6 June 2020 [1]. Since the numbers are still increasing [1], countries' policies proclaimed social isolation as an effective measure of combating the pandemic [2]. In line with global efforts, between 19 March and 11 May, 2020, the Croatian Government adopted measures to restrict gathering in public places and parks, suspend public transportation, and close institutions. Besides all social gatherings, work in retail and services including sports activities were prohibited. People living in cities and urban areas seem to be the most affected by these measures, as they were obligated to stay at home and likely reduce their common activities. Reducing the PA level could have a negative health impact, even among healthy, uninfected people [3].

Physical inactivity is one of the leading risk factors for developing numerous non-communicable diseases and shortening life expectancy [4], and represents a large economic burden [5]. What is more,

physical inactivity seriously affects human physiology and leads to dysfunction in glucose homeostasis, a decrease in protein synthesis, muscle deinnervation, and poorer immune function [6]. These health consequences are evident even after a few days of inactivity [6]. Besides, decreasing PA levels in youth can have serious consequences on health status in older age. Generally, several determinants influence PA behavior, including sex, age, socioeconomic status (SES), and psychosocial factors (e.g., social support, attitudes, motivation, etc.) [7,8]. Some of these factors may likely boost or attenuate the general impact of the above social and spatial restrictions on PA level. In addition, routine activities—such as attending university lessons or working, spending time active, and participating in sport—are among the factors expected to moderate the level of habitual PA over the course of the pandemic in large cities [9–13].

A number of commentaries, recommendation papers, and theoretical reviews have been written about the possible negative consequence of pandemic on one's health [3,14–16]. Experts, scientists, and world health institutions are warning of the potential negative effect on health of physical inactivity as an indirect consequence of the COVID-19 quarantine [3,14–16]. A recent review highlights the potential consequence of inactivity and sedentarism due to the pandemic on cardiovascular, metabolic, and neuromuscular health [6]. What is more, according to FitBit data, a 7% to 38% decline in step counts was noted during the week ending 22 March 2020 [17]. Moreover, the amount of experimental research on this topic is increasing [18–20]. Two recently published large-scale studies showed the negative impact of quarantine on PA levels in different populations [19,20]. In addition, several small-scale studies reported decreased PA levels in different populations after restrictions due to the COVID-19 lockdown [21–24]. However, some studies found that one section of active participants decreased their PA level, and that some inactive subjects tended to increase their PA level during lockdown [21–24]. These studies were performed on adolescents or older adults and did not examine factors that could possibly boost or attenuate (i.e., moderate) the change in the PA level in a homogenized group of young urban adults. However, more research is needed in this field to determine the potential consequences of restrictions in different populations. Therefore, this study aimed to investigate the moderators of change in PA level after 30 days (30-d) of restrictions due to the COVID-19 pandemic in young urban adults. Since the restriction measures disrupt these people's everyday routines to the greatest extent, any change in their PA level could be likely affected by the presence or absence of habitual daily behaviors.

## 2. Materials and Methods

### 2.1. Participants

This is a cross-sectional study conducted in Zagreb (Croatia) as an extension of a longitudinal CRO-PALS study which was performed from 2014 to 2017. Details on the sampling and procedures of the CRO-PALS longitudinal study have been described elsewhere [25]. In brief, using stratified two-stage random sampling procedures (school level and class level), 54 classes in 14 secondary schools in Zagreb were selected to participate in the study. All 1408 students in the selected classes were approached, and 903 agreed to participate (response rate = 64%). In this way, we selected a random sample of urban adolescents. From 903 adolescents that have participated in a longitudinal CRO-PALS study, 363 participants voluntarily left their contact details for the purpose of future investigations (40.2%). These participants were contacted via e-mail on 24 April 2020. After each of the subjects received a personalized identification number, a link to a computerized on-line questionnaire was sent by e-mail. A total of 104 participants completed the questionnaire (response rate = 28.7%). The data were collected from 24 April to 8 May 2020. However, 13 participants did not regularly complete the questionnaire and were excluded from further analysis. Therefore, for the purpose of this study, the total number of subjects was 91 (65% female (n = 59, mean age ± SD = 21.6 ± 0.4); 35 male (n = 32, mean age ± SD = 21.5 ± 0.3)).

To assess response bias, we compared responders and non-responders regarding age, sex, SES, and moderate-to-vigorous physical activity (MVPA) based on data before and during the restrictions. We used a t-test to assess difference between responders and non-responders in age and MVPA. To assess the difference between responders and non-responders in the proportion of females and in the proportion of subjects in all SES items, hi-square was employed. After analysis, there was a difference between responders and non-responders regarding all variables. Responders were much older, on average, in age (18.6 y vs. 21.6 y, $p < 0.0001$). Moreover, the proportion of females who responded to questionnaire was much higher than among those who did not respond (64.8 vs. 47.6%, $p = 0.001$). Regarding socioeconomic status (SES), there was an evident difference in proportion between responders and non-responders in all items of SES, where a higher proportion of non-responders was found in the first three items of SES ($p < 0.0001$). However, a higher proportion of responders was found in last two items of SES ($p < 0.0001$). In addition, responders were more physically active daily on average (MVPA: 156.3 min/day vs. 64.3 min/day, $p < 0.0001$).

The Ethics Committee of the Faculty of Kinesiology of the University of Zagreb (Croatia) approved the procedures of the CRO-PALS study (No: 1009–2014), which was performed according to the Declaration of Helsinki.

## 2.2. Procedures

### 2.2.1. Outcome Variable: Physical Activity Level

PA level was assessed using the School Health Action, Planning, and Evaluation System (SHAPES) questionnaire [26]. The SHAPES questionnaire has previously been shown to be a valid and reliable instrument for assessing PA in primary and secondary school children [26]. The PA module of the questionnaire includes two items requesting a 7-day recall of moderate intensity physical activity (MPA) and vigorous intensity physical activity (VPA). Participants had to indicate the number of hours (0–4 h) and 15-min increments (0–45 min) that MPA and VPA were performed for each day of the previous week. VPA was defined as "jogging, team sports, fast dancing, jump-rope, and any other PA that markedly increased your heart rate and made you breathe hard and sweat", while MPA was defined as "lower intensity physical activities such as walking, and riding a bike". Average day time spent while performing moderate to vigorous PA (MVPA) was calculated by summing the weekly time spent performing VPA and MPA divided by 7. For all days at which >4 h of MPA or VPA was reported, the duration of 4:15 h was assumed. In this way, the outcome variable MVPA was assessed and calculated in order to obtain information about values during the restrictions period. This variable was coded as MVPApost.

Change in PA patterns was assessed separately with a one-item question: "In the last 30 days, since the start of the restrictions due to COVID-19 pandemic, your time spent on PA is: (1) same as before pandemic, (2) higher than before pandemic, (3) lower than before pandemic". For participants who stated that they changed their usual behavior, the participants had to indicate the number of hours (0–8 h) and 30-min increments (0–45 min) that they spent more/less on average per day. Values of the variable MVPA before restrictions were coded as MVPApre. In order to calculate the value of the MVPA before the restrictions (e.g., MVPApre), we used the following formulas: *(1)* if the participant stated that their behavior has changed in the positive direction (e.g., my PA level has increased during the restrictions), that value was subtracted from the reported value which indicated behavior after 30-d of the restrictions:

$$Value_{(before\ restrictions)} = Value_{(after\ 30\text{-}d\ of\ restrictions)} - Value_{(as\ positive\ change\ before\ restrictions)}$$

*(2)* If the participant stated that their behavior had changed in a negative direction (e.g., my PA level has decreased during the restrictions = MVPApre), that reported value was *added to* the value which indicated behavior after the 30-d of the restrictions:

$$Value_{\text{(before restrictions)}} = Value_{\text{(after 30-d of restrictions)}} + Value_{\text{(as negative change before restrictions)}}$$

### 2.2.2. Moderators: Sex, Activity Status, Student Status, and Sport Participation

Sex (female/male) was used as a moderator in a separate analysis. Activity status was determined based on MVPA values before 30-d restrictions, and the calculation of this metric (e.g., MVPApre) is described within Section 2.2.1. Participants who were in the lowest quartile of MVPA and did not participate in sport activities before the restriction measures were classified as non-active, while others were classified as active. In this way, we used activity status (physically active/non-active) as the moderator in further analysis. To assess student status, the one-item question was asked: "Your current status is: 1—Student, 2—Employed, 3—Unemployed, 4—Other". After that, a new variable was created: student status. All participants who stated that they were students were classified as such, while others were classified as non-students. This variable, student status (student/non-student), was used as a fixed factor in further analysis. The original SHAPES questionnaire was supplemented with two YES/NO questions inquiring about regular participation in organized sports before the restrictions. Therefore, we created a new variable: sports participation (yes/no).

### 2.2.3. Confounders: Age and SES

Age was recorded in one-month intervals. To assess SES, a one-item question was asked: "How would you rate your socioeconomic status?". Responses were as follows: 1—Much lower than average, 2—Lower than average, 3—Average, 4—Higher than average, 5—Much higher than average.

### 2.3. Statistical Analysis

To examine the difference between pre- and post-30-d restrictions in MVPA level, four separate repeated-measures ANOVAs were employed using sex (female/male), activity status (physically active/non-active), sports participation (yes/no), and student status (yes/no) as fixed factors. Given that the study has multiple potential moderators, we tested interaction in the different models (interactions were tested in separated ANOVAs). In the case of significant difference within groups, between groups, and interactions, Fisher's LSD post-hoc test was used. To investigate which factors contributed most to the MVPA level, after 30-d of restrictions, a multiple regression analysis was performed with the following predictors: sport participation and MVPA before restrictions. The analysis was adjusted for sex, age, and SES. Estimates (coefficient) are presented as unstandardized and noted with the beta symbol ($\beta$). Multiple regression analysis was performed by using statistical package MLwiN (version 3.04) [27] while for descriptive statistics and for repeated measures ANOVA, statistical package Statistica (version 13.5) was used. The level of the statistical significance was set at $p < 0.05$.

## 3. Results

The basic characteristics of the participants for MVPA (before and after restrictions), change in PA level, sport participation, and SES (before restrictions) are shown in Table 1.

**Table 1.** Basic characteristics of the participants.

|  |  | Female (n = 59) | Male (n =32) |
|---|---|---|---|
| **MVPA (min/day)** | **MVPApre median (IQR)** | 120.0 (227.1) | 135.0 (127.5) |
|  | **MVPApost median (IQR)** | 64.3 (75.0) | 85.7 (56.8) |
| **Change in PA level (min/day) n (%)** | **No change** | 15 (25) | 10 (31) |
|  | **Increase** | 11 (19) | 6 (19) |
|  | **Decrease** | 33 (56) | 16 (50) |
| **Sport participation n (%)** | **No** | 49 (83) | 22 (69) |
|  | **Yes** | 10 (17) | 10 (31) |
| **SES n (%)** | **1** | 0 | 0 |
|  | **2** | 3 (5) | 2 (6) |
|  | **3** | 27 (46) | 10 (31) |
|  | **4** | 24 (41) | 16 (50) |
|  | **5** | 5 (8) | 4 (13) |

MVPA: Moderate-to-vigorous Physical Activity; MVPApre (min/day): Moderate-to-Vigorous Physical Activity before restrictions; MVPApost (min/day): Moderate-to-Vigorous Physical Activity after 30-d of restrictions; (IQR): interquartile range; Change in PA level: Number (%) of participants within sex group that have changed Physical Activity level after 30-d of restrictions; Sport participation: the proportion of female/male participants within sex group before 30-d of restrictions; SES n (%): socioeconomic status expressed as the proportion of female/male participants within sex group for each SES category before 30-d restrictions (1—Much lower than average, 2—Lower than average, 3—Average, 4—Higher than average, 5—Much higher than average).

Figure 1 shows changes in MVPA between pre- and post-30-day restriction measures due to the COVID-19 pandemic in male and female, active and non-active, students and non-students, and between sport participants and non-sport participants.

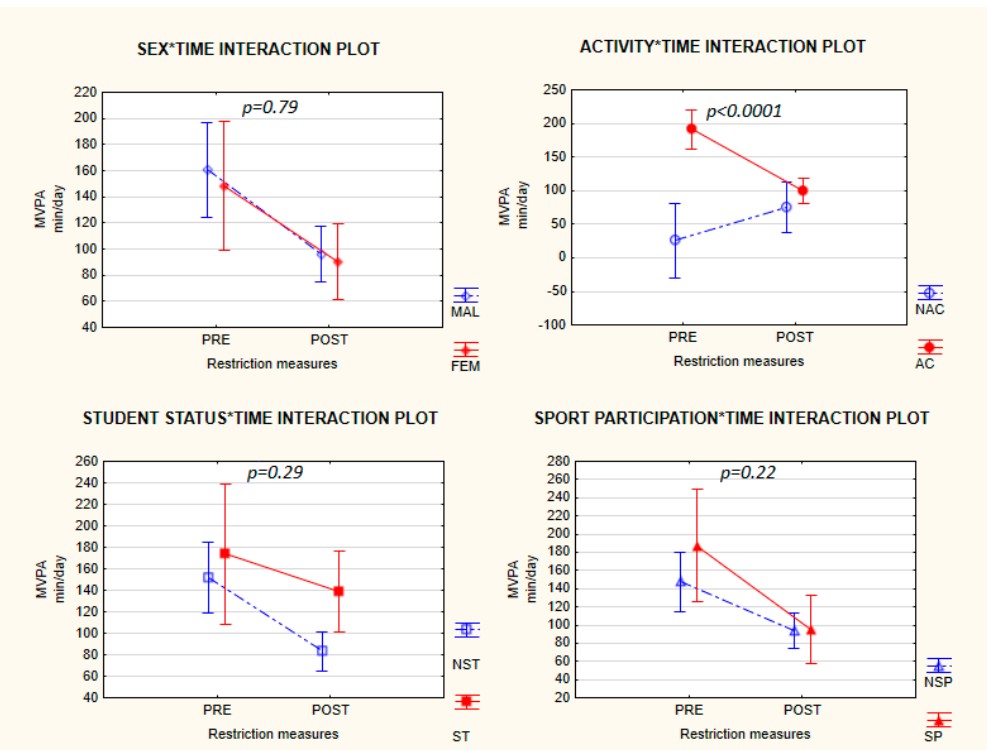

**Figure 1.** Changes in MVPA (min/day) between pre and 30-day post restrictions in female and male, active and non-active, students and non-students, and between sport participants and non-sport participants. In each interaction plot, the *p*-value represents the statistical significance for the interaction.

There was no interaction between sex and time ($F = 0.07$, $df = 1$, $p = 0.79$), MVPA level decreased among girls by 64.8 min/day ($p < 0.0001$), while in boys MVPA level dropped by 57.7 min/day ($p = 0.006$) after lockdown. Significant activity x time interaction ($F = 25.35$, $df = 1$, $p < 0.0001$) indicates that activity status before the pandemics significantly moderated MVPA level during the restriction measures. Active participants decreased their MVPA level by 100.7 min/day (within-group effect: $p < 0.0001$), while participants who were non-active before lockdown increased their MVPA level after 30-d restrictions (increase of 48.9min/day, within-group effect: $p = 0.051$). Non-significant student status x time was shown ($F = 10.51$, $df = 1$, $p = 0.29$). The within-group effect was not evident in both groups (within students MVPA significantly decreased by 69.0, $p < 0.0001$) while for non-students a non-significant reduction of 35.0 min/day was evident, $p = 0.22$). Non-significant sport participation x time interaction was shown ($F = 1.55$, $df = 1$, $p = 0.22$). In sport participants, a marked reduction in MVPA level was observed (MVPA drop by 95.3, $p = 0.001$). While in subjects who did not participate in sport before restrictions, MVPA dropped by 54 min/day; $p < 0.0001$).

Table 2 shows the results of the multiple regression analysis for MVPApost. Regression analysis revealed that MVPApre has significant and positive effect on MVPApost ($\beta = 0.297$, *S.E.* = 0.056; *95% CI* = 0.188, 0.406; $p < 0.0001$) while other predictors have not reached statistical significance ($p = 0.592$–$0.993$, see Table 2).

**Table 2.** Multiple regression analyses for MVPA after 30-d of restrictions.

|  | $\beta$ | *S.E.* | **95% CI** | *p*-Value |
|---|---|---|---|---|
| **Response** | **MVPApost** | | | |
| **Intercept** | 33.239 | 431.231 | −811.959, 878.437 | 0.939 |
| **MVPApre** | **0.297** | **0.056** | **0.188, 0.406** | **0.000** |
| **Sex** | −0.136 | 16.020 | −31.535, 31.263 | 0.993 |
| **Age** | 0.968 | 19.739 | −37.721, 39.656 | 0.961 |
| **SES** | −1.147 | 10.322 | −21.377, 19.084 | 0.912 |
| **SportPart** | −9.973 | 18.617 | −46.461, 26.514 | 0.592 |

$\beta$: Beta unstandardized regression coefficient; *S.E.*: Standard Error; *95% CI*: 95% Confidence Interval; Sex: reference category: males; SES: Socioeconomic Status; SportPart: Sport Participation, reference category: sport participants; MVPApost: Moderate-to-Vigorous Physical Activity after restrictions, MVPApre: Moderate-to-Vigorous Physical Activity before restrictions.

## 4. Discussion

This study investigated the effect of different moderators of change in PA level among young adults after 30-d of restrictions during the COVID-19 pandemic. The results suggest that restriction measures equally affected the PA of both sexes as there was an evident drop in MVPA in both female, and male young adults. On the other hand, our results indicate that restriction measures affect previously active and inactive groups of young adults differently. More specifically, previously sufficiently active people decreased their MVPA level during the restrictions, while inactive people accumulated almost 50 min/day more MVPA during lockdown on average than before the epidemic). Finally, when we compared groups of students and non-students as well as sport participants and non-sport participants, the effects of 30-d restriction measures on MVPA duration tend to be the same in all of the groups.

Unsurprisingly, our results show that, overall, PA has decreased during restrictions, a finding that has also been reported in other similar recent studies [22–24]. While other studies attributed this drop in PA to the limited ability for participation in organized sport due to restrictions imposed, the drop in MVPA in this study was similar in both youths that usually participated in organized sports and their peers that did not. On the other hand, this study indicates that prior PA status (active/non-active) is a significant moderator of change in MVPA during the 30-d of restrictions. It seems that participants who were insufficiently active prior to lockdown have managed to increase their MVPA during the

30 days of restrictions. This might be facilitated by a lower starting PA level of inactive youth as compared to their sufficiently active peers (25.0 min/day and 95.0 min/day, respectively). Nevertheless, the difficulties in introducing changes in PA behavior among the least active portion of the population are well known. Since lack of time is one of the most frequently reported barriers to exercise [28], the increase in MVPA in inactive participants might be driven by the fact that, due to movement restrictions, people had more time for other activities (e.g., exercise). It has been repeatedly shown that inactive people gain great benefits from even a modest amount of PA compared to their active peers with regards to premature mortality [29], the period of restrictions can represent a window of opportunity to introduce behavioral change to the portion of the population with the highest risk, and maximize benefits for public health. Still, an important question remains: *'Whether the acquired values of PA will be retained after the restrictions remission or will be the same and even lower as before lockdown?'* Although the answer to this question remains unanswered, we aim to conduct a follow-up study and try to gain a better understanding of this phenomenon. On the other hand, it would also be interesting to unravel the reasons why active people have difficulties in maintaining their PA level during the lockdown. We propose at least two possible explanations for this. First, sports facilities and sports parks and playgrounds were closed, which markedly reduced access to places for exercise. Second, participants who were active before the pandemic need much more physical space to engage in movement and activities, since their MVPA before the lockdown was much higher compared to their non-active peers.

Two similar studies have investigated levels of PA in active and inactive individuals, as well in male and female adults during the restrictions. Giustino et al. [22] investigated levels of PA before and during the last seven days of the COVID-19 quarantine in the Italian active population (n = 802, mean age: 32.27 ± 12.81 y.o.). They found that the number of highly active participants dropped (26%, n = 193), and the number of low and moderately active subjects increased (19%, n = 200; and 7%, n = 409, respectively). In the same study [22], both females and males decreased their total weekly energy expenditure, where males showed more reduction. The second study, performed by Lesser and Nienhuis [21] examined the influence of restrictions on PA level and well-being in the Canadian population (mean age ± SD = 42 ± 15, n = 1098). Results of this study revealed that 22% of active participants decreased their PA and 37% did not change their PA level at all. However, among inactive participants, 33% become more active and 26% did not change their PA level after restrictions. Although the mentioned results are difficult to compare with the results of our study due to different methodologies used, we can see a drop in PA level among active subjects and increase in PA among non-active participants during restrictions. A recent study by Sekulic et al. [23] examined the trends of changes in PA levels among 388 adolescents (mean age = 16 yrs) during restrictions in southern Croatia. This study found a significant drop in PA overall. However, when analyses were stratified by gender, the drop in PA level during restriction measures was seen in boys but not in girls. The findings from Sekulić et al. and Giustino et al. are not in accordance with our study considering the effect of sex on the change in PA due to restrictions. However, it is important to note that these studies used different methodologies and instruments for the assessment of PA, which severely limits comparability. In addition, the difference in populations that were studied can also potentially contribute to the difference in reported results (16 y.o. adolescents vs. 20-yo young adults vs. and 32 y.o. adults). In addition to small-scale research, two large-scale studies reported the effects of the COVID-19 lockdown on PA behavior. One recently published article examined whether the quarantine measures in the United Kingdom had disproportionate impacts on the intensity of PA in groups of people who are, or who perceive themselves to be, at higher risk from COVID-19 [20]. The results of this study showed that doing less intensive PA during the quarantine was related to obesity, hypertension, lung disease, depression, and disability. Another large-scale study was published recently and included 35 research organizations from Europe, North-Africa, Western Asia, and America [19], and showed that the COVID-19 quarantine had a negative effect on both VPA and MPA levels. In addition, many recommendations and theoretical papers have emphasized the

importance of maintaining regular PA level and physical exercise during pandemic. It is important to note that a reduced PA level can have serious negative impacts in apparently healthy individuals [6]. Therefore, exercise interventions along with a more general PA intervention need to be implemented immediately on the national level all around the world, both during and after the pandemic.

This study has several strengths. First, while most similar studies have used on-line surveys with convenience sampling, this study has investigated PA level during the COVID-19 pandemic in a subsample of a randomly selected cohort of young people, thus reducing the possibility of sample bias. Second, we considered several potential moderators of behavior change which had not been reported previously.

On the other hand, several limitations also need to be acknowledged. Firstly, PA was not measured, but was self-reported through a questionnaire, which typically leads to recall bias. Still, the SHAPES questionnaire used in this study has been previously shown to be a reasonably valid and reliable instrument for assessing PA [26]. Moreover, the same instrument was used for the assessment of the change in PA duration after restriction measures. Secondly, the low response rate and consequently relatively small sample limit the ability to generalize our results. However, the participants of this study differ from the cohort members that could not be reached, and, therefore, this study lacks generalizability.

There are several implications of this study for subsequent research, policy and practice. The study was designed as an observational cross-sectional study investigating how different moderators of change in PA affect PA level during COVID-19 lockdown in young urban adults. Follow-up studies are needed to examine the long term consequences of the changes in PA level during lockdown. This research will be expanded by a follow-up study that will include the examination of the level of PA after the dissolution of restriction measures in previously active and non-active young urban adults. The study reveals that the 30-d of restrictions have equally affected young urban adults of both genders. PA status before the COVID-19 pandemic has moderated the level of MVPA during the restriction measures. The lockdown restrictions during the early wave of the pandemic in spring 2020 have affected active and non-active young urban adults differently. In particular, more active people decrease their MVPA level, while non-active people tend to increase their MVPA level during lockdown. These findings could be useful for different stakeholders to implement measures to prevent a decrease in PA in young urban adults during the pandemic. The promotion of regular PA is of particular importance for maintaining an adequate level of PA during lockdown in physically active and inactive individuals. Moreover, health promoting measures and preventive interventions during restrictions could represent an opportunity to improve the lifestyle behavior for inactive people. Finally, our results suggest that ensuring the possibility to safely conduct outdoor activities, and adapting equipped indoor training facilities, where possible, is of particular importance during public health restriction measures, especially for more active individuals, whose exercise requires more space and equipment.

In conclusion, this study examined the effect of different moderators of change in PA during the COVID-19 lockdown in a group of young adults. While the reduction in the duration of MVPA during lockdown was similar across sex, student status and prior participation in sport, baseline activity level was shown to be a significant moderator of change in MVPA. The finding that MVPA had markedly increased in previously insufficiently active individuals during lockdown deserves to be acknowledged by public health authorities in similar circumstances in the future, when measures to reinforce this positive behavior change will need to be introduced.

**Author Contributions:** Conceptualization, J.K., M.S., I.R., and M.M.-D.; Methodology, J.K., M.S., I.R., and M.M.-D.; Validation, M.S., I.R., and M.M.-D.; Formal Analysis, J.K.; Investigation, J.K., M.S., I.R., and M.M.-D.; Resources, M.S. and M.M.-D.; Data Curation, M.S., I.R., and M.M.-D.; Writing—Original Draft Preparation, J.K.; Writing—Review and Editing, M.S., I.R., and M.M.D.; Visualization, J.K.; Supervision, M.S. and M.M.-D.; Project Administration, M.M.D. and M.S.; Funding Acquisition, M.M.D., M.S. All authors have read and agreed to the published version of the manuscript.

**Funding:** All costs of the proposed research are covered within the project *Croatian* physical activity in adolescence longitudinal study (*CRO-PALS*), which is financed by the Croatian Science Foundation (IP 2016-06-9926). The work of J.K. was funded by the Croatian Science Foundation, under the grant no: DOK-2018-01-2328.

**Acknowledgments:** The authors wish to thank to all participants for their voluntary participation.

**Conflicts of Interest:** The authors declare no conflict of interest.

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
