# Peer review of "Moderators of Change in Physical Activity Levels during Restrictions Due to COVID-19 Pandemic in Young Urban Adults"

_sustainability, doi:10.3390/su12166392_

Round 1

Reviewer 1 Report

Thanks for inviting me to review this manuscript. This study examined the effects of several potential moderators on the changes in MVPA between pre and 30-day post restrictions due to COVID-19 in youth. This is a timely study, findings of which may help people better understand the changes of their behaviors, factors for the changes and how to maintain and improve health during this uncommon situation. Please see more details about my comments and suggestions below.

Abstract:

  • The purpose of the study was not clearly stated.

Introduction

  • Lines 30-34: since the infected and death cases keep increasing and changing, it is better to specify the date by when the numbers retrieved.

Methods:

  • Have all the 903 people reported their physical activity before the restrictions? If yes, did the authors comparer the difference in physical activity between the included (104) and non-included (799) based on their data before the restrictions? Since there is small sample (N = 104), the comparison will indicate if there was any bias among the small sample.
  • In 2.2.2. Moderators sex was listed as a moderator and in 2.2.3. Confounders: Sex, Age and Socioeconomic Status, Sex was listed as a confounder, what did the authors propose the sex to be?
  • How the interactions were tested given that the study has multiple potential moderators (i.e., Sex, Activity Status, Student Status, and Sport Participation)? Did the authors test interactions in the same model or different models?

Results

  • I don’t think the terms such as “sex x time x MVPA level interaction” are correct as changes in MVPA is one of outcome variables but not one of independent variables. So, it is not appropriate to include MVPA in the interaction term.
  • Table 1. The SES is a categorical variable which has 0% for category 1 and a few for categories 2 and 5. How were they included in the models as a confounder? That is, how was the model controlling for it?
  • The p value cannot be p<.00.
  • Results of Table 2 should be reported before the interaction as those results are premise of the interaction. In other words, if the changes are not significant, there was no need to conduct the rest of the analyses.

Discussion

  • I don’t think the results (numbers) need to be re-reported in the discussion.
  • In the strengths, the authors stated “First, this study has investigated … in the homogenized population of young adults which can provide more precise conclusions.” But in the limitations, the author stated that “relatively small sample size and the population examined in this research can limit generalization.” So, is the homogenized population a strength or limitation?
  • The conclusion of the study is missing.
  • What implications are there based on the finding of this study?

Reviewer 2 Report

The paper presented for revision give us signifficant inside of the effects of pandemy on physical activitiy in young urban adults population. Introduction is clearly writteen and give us insight to the problem. Methods used are apropriate and results clearly presented. It should be mentioned that the title could be changed with added word "urban": Pandemic in urban young population since the population come from urban area and is not necessary that the same results would be obtained in rural population.

Regarding disscussion it should be pointed out that also other factors could be considered in decrease of regular PA participants. One of them could be that they participated in organised sport (clubs, schools) which was not performed (closed facilities) during pandemics. From the presented methods is not clearly visible if the data of those who participate in organised and non organised sport exists. If they exists it could be interested to present the change in PA in those who participated through organised PA and those who not. Overall presented paper give us interesting findings regarding the effects of pandemics on urban population of young adults and it could be useful to different stakeholders to implement measures during eventual successive pandemics to prevent decrease of PA in young adults.

Reviewer 3 Report

The aim of this study was to investigate moderators of change in PA level after 30 days (30-d) of restrictions due to the COVID-19 pandemic in young adults. The approach of working with an online survey during this period is appropriate. However, in order for the authors to make this suitable for publication, moderate revisions are required. There are some areas that require rewriting or clarification. I will comment on these areas section by section.

The introduction is easy to read, however did not extend existing knowledge on this topic. I suggest that the authors focus the introduction on the main goal of the study rephrasing  the initial  paragraph that is much too long (lines: 28-46). In my opinion, it is necessary to better clarify the concept of “moderators of change” and specify those of physical activity (e.g. Sex, Activity Status, Student Status, and Sport Participation). The introduction should include more update references regarding moderators of change in physical activity. Moreover, after the purpose statement, please provide a hypothesis for what the authors think the results will yield.

Methods and results

A total of 104 participants completed the questionnaire (response rate = 28.7%). However, 13 participants did not regularly complete the questionnaire and were excluded from further analysis. Therefore, for the purpose of this study, the total number of subjects was (65 % 92 female (n = 59, mean age ± SD = 21.6 ±0.4); 35 % male (n = 32, mean age ± SD = 21.5 ±0.3)). The small simple size is an important limitation for the  generalizability of the results.

Discussion:

It is not surprising that Covid-19 confinement reduced physical activity levels in active people. However, it has also been shown that non-active people increased their PA level during lockdown. Restrictions for them can represent an opportunity to change their behavior in a positive direction in order to gain better health status. Please better discuss these interesting results in discussion.

The authors failed to discuss the limitations of the study. There are several large drawbacks inherent to the study design and implications of theses on the generalizability of the results needs to be discussed.

Please better list strengths and limitations of the study. Consider all the biases inherent to the study design. An important limitation of this study was self-reported data by parents, which may lead to recall bias. Also, due to the convenience sampling method using online platforms, it may be a selection bias. Please better discuss this limit.

Round 2

Reviewer 1 Report

Abstract:

Point 1: The purpose of the study was not clearly stated.

Response 1: Thank you for this comment. We corrected first sentence in the abstract about purpose of this study (p.1., line 10.-11.).

R1 comment: Thanks!

Introduction:

Point 2: Lines 30-34: since the infected and death cases keep increasing and changing, it is better to specify the date by when the numbers retrieved.

Response 2: Thank you for provided suggestion. We integrated exact date into the text (p.1, line 30).

R1 comment: Thanks!

Methods:

Point 3: Have all the 903 people reported their physical activity before the restrictions? If yes, did the authors comparer the difference in physical activity between the included (104) and non-included (799) based on their data before the restrictions? Since there is small sample (N = 104), the comparison will indicate if there was any bias among the small sample.

Response 3: Thank you for spotting this mistake. We made additional analysis and integrated their results into text as separated paragraph under the ‘2. Material and methods’, section 2.1. Participants (p.3., line 89.-101.).

R1 comment: Thanks!

Point 4: In 2.2.2. Moderators sex was listed as a moderator and in 2.2.3. Confounders: Sex, Age and Socioeconomic Status, Sex was listed as a confounder, what did the authors propose the sex to be?

Response 4: Thank you for spotting the mistake. Sex was a moderator so we omitted it from the confounders section (p.4, line 156.-160.).

R1 comment: Thanks, but Sex is still included in the 2.2.3 heading!

Point 5: How the interactions were tested given that the study has multiple potential moderators (i.e., Sex, Activity Status, Student Status, and Sport Participation)? Did the authors test interactions in the same model or different models?

Response 5: Thank you for this comment. Given that the study has multiple potential moderators, we tested interaction in the different models (interactions were tested in separated ANOVAs).

R1 comment: Thanks, but it should be claimed in the statistical analysis section!

Results

Point 6: I don’t think the terms such as “sex x time x MVPA level interaction” are correct as changes in MVPA is one of outcome variables but not one of independent variables. So, it is not appropriate to include MVPA in the interaction term.

Response 6: Thank you for spotting this mistake. We made correction through the text and used term such as ‘sex x time interaction’ instead of previous one (sex x time x MVPA level interaction) (p.6., line 196.; p.6., line 198.; p.6., line 202. – 203.; p.6., line 205. - 206.).

R1 comment: Thanks!

Point 7: Table 1. The SES is a categorical variable which has 0% for category 1 and a few for categories 2 and 5. How were they included in the models as a confounder? That is, how was the model controlling for it?

Response 7: Thank you for this comment. We included SES as an ordinal variable. Also, SES were assessed as the ordinal variable and as such integrated in the all models.

R1 comment: Thanks!

Point 8: The p value cannot be p<.00.

Response 8: Thank you for spotting this mistake. We made correction through text (p.6., line 198.; p.6., line 200. - 201.; p.6., line 204.; p.6., line 208.; p.7., line 211.).

R1 comment: Thanks!

Point 9: Results of Table 2 should be reported before the interaction as those results are premise of the interaction. In other words, if the changes are not significant, there was no need to conduct the rest of the analyses.

Response 9: Thank you for this comment. However, we are not sure we understand what was meant by this. Namely, Table 2 presents results of the multiple regression analysis aimed at uncovering the predictors of the PA level during restriction period. Interaction terms were not entered into these models. On the other hand, interaction results presented in line 188.-200. stem from a series of repeated measures ANOVA analyses that were supposed to compare the change in MVPA among the different groups of participants. These interactions were pre-defined and are presented simultaneously with respective within-group differences. Thus, we believe that the current outline of the results is not flawed. If you still disagree, and would be happy to offer a more detailed explanation, we would be glad to follow the more detailed explanation

R1 comment: Thanks!

Discussion

Point 10: I don’t think the results (numbers) need to be re-reported in the discussion.

Response 10: Suggestion accepted. In the revised manuscript we reworded this section of the discussion and omitted the numbers (please see p.7., line 220. -229.).

R1 comment: Thanks!

Point 11: In the strengths, the authors stated “First, this study has investigated … in the homogenized population of young adults which can provide more precise conclusions.” But in the limitations, the author stated that “relatively small sample size and the population examined in this research can limit generalization.” So, is the homogenized population a strength or limitation?

Response 11: In response to this comment as well as to comments from the other reviewer we rewrote the whole strengths and limitations section. (please see p.9, line 291.-302.).

R1 comment: Thanks!

Point 12: The conclusion of the study is missing.

Response 12: Thank you for this comment. In the revised manuscript we included a concluding paragraph (p.9, line 303. – 309.)

R1 comment: Thanks!

Point 13: What implications are there based on the finding of this study?

Response 13: Thank you for this important note. We included a prolonged section on the implications within the new concluding paragraph that was added at the end of the manuscript (p. 9., line 310. – 329.).

R1 comment: Thanks! If it is not required by the journal, I suggest switching conclusion and implication sections, and putting the conclusion at the end of the paper.

Reviewer 2 Report

The presented paper with improvements is giving significant data regarding PA of young urban adults during COVID19. I would like to suggest to the authors due to the situation to follow the situation also in the case of second wave. It could be also interesting to see what is happening with young adults in rural areas. Paper can be accepted in present form.

Author Response

Dear Reviewer,

Thank you for your comments which contributed to our manuscript. Also, we wish to thank you for the suggestion and we will try to make another study with suggested comments!

Reviewer 3 Report

In my opinion the paper can be accepted in present form.

Author Response

Dear Reviewer,

Thank you for the provided comments and suggestions. Your comments improved our manuscript!